# Metaproteomics—An Advantageous Option in Studies of Host-Microbiota Interaction

**DOI:** 10.3390/microorganisms9050980

**Published:** 2021-04-30

**Authors:** Oleg Karaduta, Zeljko Dvanajscak, Boris Zybailov

**Affiliations:** 1Department of Biochemistry and Molecular Biology, UAMS, Little Rock, AR 72205, USA; BLZybaylov@uams.edu; 2Arkana Laboratories, Little Rock, AR 72211, USA; zeljko.dvanajscak@arkanalabs.com

**Keywords:** metaproteomics, metagenomics, microbiome, microbiota, dietary change

## Abstract

Gut microbiome contributes to host health by maintaining homeostasis, increasing digestive efficiency, and facilitating the development of the immune system. Manipulating gut microbiota is being recognized as a therapeutic target to manage various chronic diseases. The therapeutic manipulation of the intestinal microbiome is achieved through diet modification, the administration of prebiotics, probiotics, or antibiotics, and more recently, fecal microbiome transplantation (FMT). In this opinion paper, we give a perspective on the current status of application of multi-omics technologies in the analysis of host-microbiota interactions. The aim of this paper was to highlight the strengths of metaproteomics, which integrates with and often relies on other approaches.

## 1. Introduction

The role of the gut microbiome in human health and disease is becoming clearer thanks to high throughput sequencing technologies (HTS). The development and application of fast and low-cost DNA sequencing methods has been revolutionary. HTS has been widely used to examine the complexity of the gut microbiome due to the speed, scale and precision of information yielded. For compositional analysis, the 16S rRNA gene has been most frequently targeted due to its presence in all prokaryotes and the existence of variable domains that allow different taxa to be distinguished. While 16S rRNA studies provide data in relation to the microbial composition of an ecosystem, these do not provide direct information regarding the microbial viability or the functional potential of the populations present.

Metagenomic (or shotgun sequencing) studies go beyond the 16S rRNA gene to characterize the full genetic content of a community, thereby providing an insight into the potential functional capacity of the microbes present [1,2,3]. Regardless of the approach taken, it is important to note that these sequencing technologies require detailed bioinformatic analyses to deal with the large volumes of data generated [4,5].

Although these gene-centric approaches have provided much information regarding the content of the gut, we also need to understand the activity of these genes and their impact on the metabolic networks within the gut. To further determine specific microbial activity, it is necessary to analyze gene expression (metatranscriptomics), protein products (metaproteomics) and metabolic profiles (metabolomics). These techniques are complex, and to different extents, are still somewhat in their infancy. To date, metatranscriptomics, based on the large-scale sequencing of 16S rRNA transcripts, has been used to look at the composition of the active microbiota in healthy individuals and has revealed that the transcriptional profile across individuals is more similar than indicated by the associated taxonomic diversity [6]. Functionality between the individuals is more similar than the taxonomy, also when analyzing metaproteomics [7].

The reason that metaproteomics has become an important tool can be partially explained by the “quantum leaps” in enabling technologies that have made it more feasible and affordable. Technological advances have mostly occurred in the realm of liquid chromatography (LC) enabling the separation of highly complex peptide mixtures, and high-resolution mass spectrometry (MS) instrumentation enabling the acquisition of large numbers of accurate mass spectra, and computational tools for data processing and analyses.

## 2. Tasks That Can Be Solved through Metaproteomics

In its basic form, metaproteomics allows us to study the presence and abundances of proteins in any microbial community. With a well-curated protein sequence database in hand, we can assign these proteins to individual species or higher taxa and understand the functional roles and interactions of individual members in the community. Since proteins convey structure and activities to cells, knowing their abundances provides a picture of the cellular phenotypes at the molecular level. Pure culture studies from a diversity of organisms have shown that the abundance of a protein indicates its relevance and activity under a given condition. Applying this at the community level can lead to critical insights and significant progress. For example, in a marine worm that relies on its five bacterial symbionts for all nutrition, we did not know which environmental energy sources the symbionts were using for carbon fixation. A metaproteomic study revealed that some of the symbionts abundantly expressed carbon monoxide dehydrogenases, which suggested they were using carbon monoxide as an energy source [8].

Differential metaproteomics can be used to identify changes in the expression of individual genes, for example, after actively manipulating a microbial community by changing substrates or if environmental conditions naturally fluctuate. In a recent publication, studies have used this approach to show that hydrogen transfer is the basis of the symbiosis between a Breviatea protist and its Arcobacter symbiont by comparing the metaproteomes from both partners to those of only one partner [9,10].

## 3. Novel Software Tools in Metaproteomics Analysis

Most of the reported metaproteomics studies use shotgun proteomics for the identification of bacterial proteins, which may be influenced by complexity and sensitivity issues, such that the detection of low abundant proteins may be challenging. However, the complexity and dynamic range issues may be addressed using the emerging spectral library-based methods such as data-independent acquisition (DIA), which provides a library-based, broader coverage for peptide/protein detection [11,12,13,14,15]. While such an approach has been increasingly applied in the quantitative analysis of the human proteome, its applications in metaproteomics has lagged behind, in part due to the complexity involved in bioinformatics. Reference databases are constantly integrating mass spec analysis of cultured bacterial and pathogen species to generate reference databases for improved species identification [16,17]. Recently developed software tools specifically designed for metaproteomic data analysis have become available as well. MetaLab uses spectral clustering to improve peptide identification speeds [18]. Others have shown improved protein identification in metaproteomics by employing de Bruijn graph assembly to predict protein sequences from metagenomics sequence data and generate a reference database [19,20]. The taxon-specific classification of peptide sequences can be performed using UniPept (http://unipept.ugent.be) which uses shotgun proteomics data from UniProt KB with identification noise filtering to provide enhanced biodiversity analysis [21,22]. Another option for the analysis of metaproteomics data is MetaPro-IQ, which is ideal for fecal samples since the gut microbiome gene catalog was curated from fecal studies which negates the need for matched metagenomics data but makes it less applicable for other types of microbiome samples [21].

## 4. Metaproteomics Pipeline

In recent years, progress in developing a novel technique enabling the species-level resolution of gut microbiota has been shown. Corresponding data analyses usually consist of numerous manual steps that must be closely synchronized. The general approach usually consists of getting raw mass spectrometry data, processing it with sequential steps of database construction, peptide identification and quantification, taxonomy and function profile construction. A variety of methods and platforms are utilized to ease this process, such as iMetaLab [23], MetaProteomeAnalyzer and Prophane [24].

In our laboratory, we developed metaproteomic methods to study animal- and human-associated microbiota with a focus on understanding the critical interactions between all partners, including the host (Figure 1). Such studies are challenging for multiple reasons, including the presence of fecal or soil-derived substances that interfere with sample preparation and the high abundances of host-associated proteins that potentially swamp out the microbial signal. This particular multi-step database search strategy was used in PEAKS Studio (Bioinformatics Solutions) to arrive at the final list of identified proteins [25].

At the first step, PEAKS generates de novo peptide sequence tags from the raw mass spectra without the use of any sequence database. For the initial quality assessment of metaproteomics samples, the list of de novo peptide sequences was supplied to UniPept, an online metaproteomics tool [22,26].

At this initial step, one can assess the quality of sample preparation by comparing ratios between host and bacterial proteins, with more bacterial proteins indicating higher quality. Outlier samples, where the bacterial proteins are under-represented or missing, can be removed at this step, to save time in the downstream analysis. At the second step, the PEAKS-generated de novo tags are searched against large sequence databases, which include both host and bacterial proteins. These databases have millions of entries, and after the list of identified proteins is obtained, their sequences are further used to create a custom, smaller database, against which the lists of de novo peptide tags are searched again, to arrive at a final protein list.

For protein quantification, Scaffold v.4 with the quantitation module (Proteome Software) was used. Data and the custom FASTA database were exported from PEAKS into Scaffold as mzIdentML and mascot generic format files. Scaffold-derived normalized spectrum abundance factors (NSAF) values and Total Unique spectral counts were used for the statistical analysis in R/bioconductor [27]. Taxonomic information from the NCBI-nr and Uniprot database was used to derive taxonomic units. The sum of spectral counts matching a given taxonomic unit, weighted by a total spectral count per a given replicate, was used as a measure of abundance for that taxonomic unit. Moderated t-test (limma package, Bioconductor) was used to establish the differential abundance of taxonomic units. Bonferroni-Hochberg correction for multiple testing was enforced for both protein and taxonomic unit quantification [28].

This method might improve our understanding of the host-bacteria interaction with further progress not only in disease management; it might improve and deepen our understanding of how neonatal diet alters fecal microbiota profiles. Even though a massive study to investigate fecal microbiota and metabolites at different ages in infants who were breastfed, received dairy-based milk formula, or received soy-based formula was performed, authors were not able to capture species-level differences due to the limitations of 16S rRNA sequencing [29].

This method could be also be a part of a multi-omics approach in studies aiming to elucidate longitudinal dynamics in the microbiome structure [30] or in studies of modulating gut microbiota composition in a preclinical neonatal model [31].

## 5. Technical Challenges and Future Perspectives

Despite metaproteomics having been around for fifteen years, it is still in its infancy and there is great potential for its continued development to address otherwise intractable questions. The appearance of new technologies, training additional personnel on metaproteomics approaches, validating and standardizing approaches, developing custom computational tools, and establishing dedicated metaproteomics meetings are the keys that will drive this method forwards.

Predictions about the future of metaproteomics need to anticipate its future applications. Foreseeable trends are an increase in MS resolution and therefore more data will be acquired. Although metaproteomics is a very powerful method, issues within the bioinformatics evaluation impede its success. There will be more studies of gut microbial protein expression in response to dietary patterns that would lead to new heights in understanding microbiome behavior [32].

Protein identification is difficult if the taxonomic composition is unknown or protein entries are missing from protein databases. For example, the UniProt/TrEMBL database only contains proteins from 1,240,268 species (http://www.ebi.ac.uk/uniprot/TrEMBLstats, status, accessed on 29 March 2021), but the number of microbial species on Earth is estimated to be up to one trillion [33]. Thereby, already small changes in the protein sequence between related microorganisms have a big impact on protein identification. One mutation in every tenth amino acid leads to completely different tryptic peptides which hinder the identification of any peptide for the investigated protein.

Another often neglected aspect is the reproducibility of results using different metaproteomics software tools. Thus far, only Tanca et al. [34] tested their complete metaproteomics workflow for a defined mixed culture of nine different microorganisms. A comparison where multiple research groups evaluate an identical sample would also be desirable. Additionally, it is evident that sample complexity and inter-individual variation in gut microbiota are extensive [35]. It is also important to note that host and microbiota interactions involve delicate interplay between factors such as age, genetics, immunity and dietary habits, which could act as potential confounding variables. Establishing an integrated multi-omics protocol will also be beneficial for the development of metaproteomics as either a stand-alone approach or as part of a complex strategy [30].

## 6. Conclusions

With the growing interest in understanding the link between gut microbiota in health and diseases, metaproteomics analysis is instrumental to characterize the activity and functional pathways of the microbial community. It represents a powerful tool for the taxonomic and functional characterization of complex microbial communities from all the variety of samples. In the future, it has the potential to become a valuable tool for routine diagnostics, e.g., the analysis of human feces. Although challenging, further advances in sample preparation methods, the development of more sophisticated analytical tools in addition to the availability of relevant software and databases are expected to facilitate the progress of metaproteomics in the coming years. Metaproteomics studies will benefit from software supporting the taxonomic and functional interpretation of results. Even if it is obvious, the close cooperation of bioinformaticians and biologists should also be considered during software development.

By harnessing the power of the emerging technologies, it is anticipated that more details on the microbial functionality and their connection with a human host will be uncovered in the future.

## Figures and Tables

**Figure 1 microorganisms-09-00980-f001:**
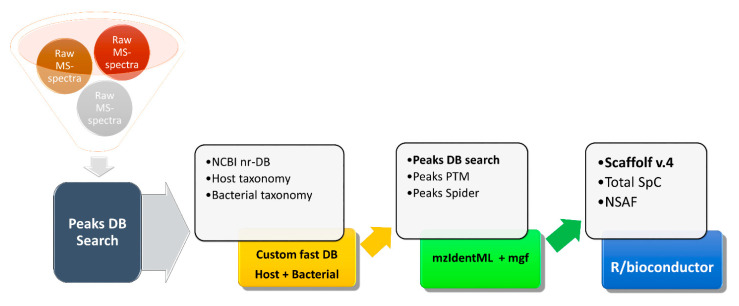
Experimental and analytical workflow. The metaproteomics data-analysis shows the software tools used to arrive at the identified proteins list.

## Data Availability

Not applicable.

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
