# Peer review of "Metaproteomics—An Advantageous Option in Studies of Host-Microbiota Interaction"

_microorganisms, 2021, doi:10.3390/microorganisms9050980_

Round 1

Reviewer 1 Report

The reviewed manuscript reads a lot better and most of the previous questions have been addressed. A few minor things to need to be changed as follows

Specific comments to revised version

I suggest to change the abstract line

 The aim of this paper is not to convince the reader  that metaproteomics is the best or only approach to study what transpires in microbial communities, 16 but to highlight the strengths of metaproteomics, which integrates with and often relies on other approaches.

L26 precision ‘of’ information

L85 space after references

L109 and 115 italicize de novo

112 change indication to indicating

Author Response

Dear Reviewer (1),

We appreciate that you found time to look through the revised manuscript. We incorporated all your suggestions throughout the manuscript, which made it much easier to read.

Thank you!

Reviewer 2 Report

Dear Authors,

I have read your manuscript and while I think the subject is very interesting I do think this manuscript could be elaborated bit more and included some more recent references.

I would also recommend some English proof reading to make the text more understandable.

Line 26 – Instead of „...precision information yielded.“  change to „precise information yielded.“

Line 41 – Instead of „...These techniques can be complex ..“ change to „These techniques are complex...“

Lines 46-47 – I don’t understand the connection between this sentence and previous. It needs to be pointed out that authors are trying to say that the functionality between the individuals is more similar than the taxonomy also when analysing metaproteomics. There is also good reference to the metabolites: Visconti, A.; Le Roy, C. I.; Rosa, F.; Rossi, N.; Martin, T. C.; Mohney, R. P.; Li, W.; de Rinaldis, E.; Bell, J. T.; Venter, J. C.; et al. Interplay between the Human Gut Microbiome and Host Metabolism. Nat. Commun. 2019, 10 (1).

I suggest all this section (lines  42-47) should be separate and elaborated more in this topic. While line 41-42 sentence could be followed to elaborate bit more on topic how those methods are complex and in their infancy.

Section “1. Tasks that can be solved through metaproteomics” is bit limited, more examples could be pointed out in the fields of metaproteomics,

Lines 80-82 – There is a recent paper where DIA is applied on metaproteomics together with development of a software.  Aakko, J.; Pietilä, S.; Suomi, T.; Mahmoudian, M.; Toivonen, R.; Kouvonen, P.; Rokka, A.; Hänninen, A.; Elo, L. L. Data-Independent Acquisition Mass Spectrometry in Metaproteomics of Gut Microbiota - Implementation and Computational Analysis. J. Proteome Res. 2020, 19 (1), 432–436.

Lines 92-96 – Very long sentence which makes the meaning bit unclear. Also, the reference is wrong.

Lines 98-99 – Instead of “…was shown” use “…has been shown” or rephrase the sentence.

Lines 156-157 – There is a recent publication (preprint) that evaluates the metaproteomics analysis of the same sample between multiple laboratories.  Critical Assessment of Metaproteome Investigation (CAMPI): A Multi-Lab Comparison of Established Workflows. doi: https://doi.org/10.1101/2021.03.05.433915

Author Response

Dear Reviewer (2),
Thank you for your detailed response!
We did our best to address all your corrections (all highlighted in red) regarding proper English.
In addition, we found recommended articles appropriate and well suited for the manuscript.
In details:

Line 26 – Instead of „...precision information yielded.“  change to „precise information yielded.“
Corrected

Line 41 – Instead of „...These techniques can be complex ..“ change to „These techniques are complex...“
Corrected

Lines 46-47 – I don’t understand the connection between this sentence and previous.
This sentence was truly an outlier. Deleted.

It needs to be pointed out that authors are trying to say that the functionality between the individuals is more similar than the taxonomy also when analysing metaproteomics. There is also good reference to the metabolites: Visconti, A.; Le Roy, C. I.; Rosa, F.; Rossi, N.; Martin, T. C.; Mohney, R. P.; Li, W.; de Rinaldis, E.; Bell, J. T.; Venter, J. C.; et al. Interplay between the Human Gut Microbiome and Host Metabolism. Nat. Commun. 2019, 10 (1).

That is a good clarification. We used this reference

I suggest all this section (lines 42-47) should be separate and elaborated more in this topic. While line 41-42 sentence could be followed to elaborate bit more on topic how those methods are complex and in their infancy.

Even though this remark looks more than reasonable, we would prefer to stick with the format of Opinion Paper, rather than return to a complex format of Review Paper. However, of course, if Guest Editor would find it vitally necessary, we will update this paragraph with more information and details. 

Lines 80-82 – There is a recent paper where DIA is applied on metaproteomics together with development of a software.  Aakko, J.; Pietilä, S.; Suomi, T.; Mahmoudian, M.; Toivonen, R.; Kouvonen, P.; Rokka, A.; Hänninen, A.; Elo, L. L. Data-Independent Acquisition Mass Spectrometry in Metaproteomics of Gut Microbiota - Implementation and Computational Analysis. J. Proteome Res. 2020, 19 (1), 432–436.
Thank you for suggestion, we included this reference as well

Lines 92-96 – Very long sentence which makes the meaning bit unclear. Also, the reference is wrong.
Corrected

Lines 98-99 – Instead of “…was shown” use “…has been shown” or rephrase the sentence.
Corrected

Lines 156-157 – There is a recent publication (preprint) that evaluates the metaproteomics analysis of the same sample between multiple laboratories.  Critical Assessment of Metaproteome Investigation (CAMPI): A Multi-Lab Comparison of Established Workflows. doi: https://doi.org/10.1101/2021.03.05.433915

We will be more than happy to utilize this information and cite this paper if respected Editor confirms that preprints are acceptable.

Reviewer 3 Report

The manuscript by Oleg Karaduta et al discussed the application of metaproteomics to study host- microbiota Interaction. Overall, the topic is interesting and the manuscript provides some nice discussion on the current status, challenges, and future perspective of metaproteomics.  There are two concerns:

  1. In section 3 Metaproteomics Pipeline, there authors only discussed the pipeline from one platform, PEAKS Studio, which was developed from the authors’ lab. This section should not be limited to one pipeline. The authors should expand the section to discuss the general approach, and examples of commonly used platforms. Alternatively, if the authors wished only to discuss the PEAKS Studio, then the title of this manuscript should reflect that.
  2. It would be more appealing to add a diagram about the metaproteomics approach.

Author Response

Dear Reviewer (3),

1) Thank you for pointing this out. We agree that it is important to expand this section and not to limit it to one pipeline. Therefore, we have not only expanded this section with description of general approach, and examples of commonly used platforms, but, since this special issue is focused on infants, we included information from several recent papers (such as https://doi.org/10.3390/microorganisms9050884; https://doi.org/10.1093/ajcn/nqaa076; https://doi.org/10.3390/microorganisms8121887).

2) We also agree that it would be beneficial for reader to have a diagram, so we included it in the manuscript. We appreciate your attention to details!

This manuscript is a resubmission of an earlier submission. The following is a list of the peer review reports and author responses from that submission.

Round 1

Reviewer 1 Report

This manuscript is a mini-review that discusses the relationship of the gut microbiome with disease and microbiome targeting therapeutic strategies. The contributions of ‘omics’ technologies to discoveries in this field are then discussed. The authors assert that mass spectrometry based metaproteomics has the most potential but need to strengthen their arguments to back this.

Major comments

This needs a short introduction tying all the ideas together and giving a roadmap for the discussion.

The aim needs to be more clear from the beginning. Line 17-19 Line starting ‘In this mini-review…. States you are summarizing ‘omic’s technologies in relation to host-microbiota interactions. This seems to be the aim but you need to include mention of human diseases as stated in the title and also that you are summarizing microbiome in relation to disease, therapies and interventions.

Lines 34-37 The last line of a section in bold states ‘the utility of this animal model for drug trials targeting gut microbiota is discussed’ Neither this animal model or drug trials are discussed. As the section is in bold is seems like it should be the hypothesis or discussion topic, however it does not represent the rest of the discussion.

The flaws of proteomics need to be discussed alongside the benefits. There are reasons this is the least used of the ‘omics’ techniques at the present even if it has great potential.

Metranscriptomics is not just of the 16S rRNA of active community members. 16S depletion and mRNA transcriptomics is also common and this does give a lot of information on functional microbiota. You need to consider the advantages/disadvantages of this method vs proteomics

Minor comments

Title - as ‘an’ advantageous ….

Line 62 change unmber to number

Line 184 change to ‘In recent years …’

Line 170 the reference databases are still far behind 16S

Give examples of where proteomics found better associations than other ‘omics’

Can you give an example of the % identification of targets in metaproteomics studies vs metagenomics

Can you give an indication of the costs of metaproteomics vs other ‘omics’

 Line 196 ‘This method might very well revolutionize chronic kidney disease management’. How?

Reviewer 2 Report

This is an interesting mini-review that aims at discussing a novel approach to study complex host/microbiome interactions. However, information presented fails to do so in detail, and includes information that is not directly linked to the aim of the paper. Details below to clarify this comment:

Introduction - there should be an introduction that focuses on what we know and what are the gaps - numbered subtitles should be avoided. The first section seems out of place for this introduction, since the title implies the focus of this review is metaproteomics. 

Section 2, first two paragraphs need referencing. This whole section does not really flow as each paragraph seems to talk about different unrelated concepts. Some of the evidence presented relates to animal studies, where human evidence is available for the same outcomes/findings.

Section 3. Bean is not a form of resistant starch is a legume.

Section 4 - references needed in first paragraph. From this section onwards information is really relevant to title. What is presented in previous sections seems disjointed and does not add value to the paper.

Section 4 and5 could be expanded to better explain the details of this techniques, and provide values of their accuracy/repeatibility, costs, etc. and how these compare with current approaches. Also authors could discuss how results from studies varies using these techniques, perhaps including studies that have used more than one method or studies that have compared methods.

Only sections 5-6-7 have the relevant information for this paper.'s aims. Yet, information is superficial and presents no evidence on metaproteomics superiority as a method to study host/microbiome interactions. There is information lacking regarding targets for metaproteomics, what kind of information one can obtain from this that cannot be obtained with metagenomics, etc. Authors state that current data base is limited and how a common mutation could hinder protein identification - these seem like big challenges that undermine validity/reliability of metaproteomics.